# A Light Shed on Lepton Flavor Universality in B Decays

**Sonali Patnaik [1,\*]** and **Rajeev Singh [2,3]**

1    Department of Physics, College of Basic Science and Humanities, Odisha University of Agriculture & Technology, Bhubaneswar 751003, India

2    Center for Nuclear Theory, Department of Physics and Astronomy, Stony Brook University, Stony Brook, NY 11794-3800, USA

3    Institute of Nuclear Physics Polish Academy of Sciences, 31-342 Cracow, Poland

*    Correspondence: patnaiksonali.29@gmail.com

**Abstract:** Behind succeeding measurements of anomalies in semileptonic decays at LHCb and several collider experiments hinting at the possible violation of lepton flavor universality, we undertake a concise review of theoretical foundations of the tree- and loop-level $b$-hadron decays, $b \to cl\nu_l$ and $b \to sl^+l^-$ along with experimental environments. We revisit the world averages for $R_{D(D^*)}$, $R_{K(K^*)}$, $R_{J/\psi}$, and $R_{\eta_c}$, for the semileptonic transitions and provide results within the framework of the relativistic independent quark model in addition to the results from model-independent studies. If the ongoing evaluation of the data of LHC Run 2 confirms the measurements of Run 1, then the statistical significance of the effect in each decay channel is likely to reach $5\,\sigma$. A confirmation of these measurements would soon turn out to be the first remarkable observation of physics beyond the Standard Model, providing a wider outlook on the understanding of new physics.

**Keywords:** new physics; B meson; B decays; LFUV; LFU ratios; anomalies; flavor physics; semileptonic transitions; LHCb; Belle; scattering amplitudes; form factors; BSM





## 1. Introduction

Within the framework of the most elegant and concise theory of particle physics, the Standard Model (SM), the basic elementary units are fermions which are further classified into quarks and leptons. For detailed properties of elementary particles, we direct the reader to Ref. [1]. So far, the leptons are considered to be point-like particles with no substructure. Leptons undergo both weak and electromagnetic interactions, whereas neutrinos participate only in weak interactions. The SM treats these charged leptons ($e^-$, $\mu^-$, $\tau^-$) to be universal, i.e., same for the three generations except for kinematic effects due to their different masses. It predicts electroweak interaction to have the same amplitude for all three different lepton generations except for phase–space differences or helicity suppression effects. This property is called Lepton Flavor Universality (LFU) [2–5], and it has been experimentally verified in $\mu$ decays [6–8], $\tau$ decays [1], and Z boson decays [9]. LFU is an accidental symmetry of the SM where the strength of Yukawa interactions between lepton–Higgs gives rise to the different lepton masses ($m_\tau > m_\mu > m_e$). This means that the physical processes involving charged leptons should feature an LFU, which is an approximate lepton flavor symmetry among physical observables, such as decay rates or scattering cross-sections. It is broken in the SM only by charged lepton mass terms. The transitions due to flavor-changing charged currents (FCCC) are only mediated by charged weak bosons $W^\pm$, and the transitions due to flavor-changing neutral current (FCNC) are mediated by neutral weak boson ($Z^0$) in the tree-level and by virtual $W^\pm$ bosons and virtual quarks in the loop processes.

However, in the current parlance of the literature, evidence of breaking of the LFU has been witnessed in $b \to sl^+l^-$ ($l = e, \mu$) using several collider experiments [10] that have shaken the fundamental understanding of physics. This means there may be some new

particles in extensions to the SM that may violate this symmetry resulting in observable changes in the rates of quark–lepton transitions in the *B*-hadron decays of the SM. Also, there has been a fascinating set of anomalies looming in the past few years in various studies of different decay modes [11–22], see Refs. [23–29] for latest theoretical investigations. First-order decays of beauty (*B*) mesons to final $\tau$ lepton states have also been observed in BaBar [30,31], Belle [11,32–35], and LHCb [10,19,21,36–40] experiments. In all cases, indications of LFU violation (LFUV) were announced. Recently, leptonic and semileptonic decays of *B* meson ($B \to \tau\nu_\tau$, and $B \to D(D^*)l\,\nu_l$ with $l = e$, $\mu$, or $\tau$) seem to challenge lepton universality. Any evidence and observation of this violation and clarification of interactions of new particles would be an open window to further explore the presence of physics beyond the Standard Model (BSM). This will provide an indirect portal to the resolutions of the nature of dark matter, the origins of the matter–antimatter asymmetry, or the dynamics of the electroweak scale.

In this paper, we present a brief review focusing on the rich anomalies in semileptonic transitions. The rates of *B*-decaying to $\tau$ and $\mu$ leptons are expected to differ because of the substantial $\mu$–$\tau$ mass difference. The charge-current anomalies involving $B \to D, D^*l\bar{\nu}_l$ decays were first measured at BaBar Collaboration in 2012 [30]. In particular, the LFU ratio is defined as:

$$R_{D^{(*)}} = \frac{Br(B \to D^{(*)}\tau\bar{\nu}_\tau)}{Br(B \to D^{(*)}l\bar{\nu}_l)}, \quad \text{with} \quad \ell = \mu, e, \tag{1}$$

where $D^{(*)}$ refers to $D(D^*)$ meson and *Br* indicates the branching ratio. The results presented in Ref. [30] disagree with the SM at 3.4 $\sigma$. These measurements were repeated again at BaBar (2013) [31], followed by Belle (2015) [32] and LHCb (2015) [36], and the results were largely confirmed. Recently, an updated measurement of $R_D$ and $R_{D^*}$ has been announced by LHCb (first joint measurement) [41] based on LHC Run 1 data which shows agreement with the previous measurement [42–44] in 3 $\sigma$ level, see Table 1 for the summary. The $\tau$ is reconstructed in $\tau \to \mu\nu\bar{\nu}$, and the result supersedes the previous result obtained in 2015 [36]. A key feature of the deviation is that the measured observables are always higher as compared to the SM predictions and thus imply LFUV. However, recently the world average has been coming closer to the SM predictions [45,46], and the significance of the deviation is still more than 3 $\sigma$ due to the reduced uncertainties. These theoretical and experimental measurements will further lead to a more dedicated parameterization of form factors and their correct determination in semileptonic transitions.

Apart from these results, there are additional measurements for various other $b \to cl\nu_l$ decays. LHCb performed another test for LFU ratio with a different spectator quark, i.e.,

$$R_{J/\psi} = \frac{Br(B_c^+ \to J/\psi\tau^+\nu_\tau)}{Br(B_c^+ \to J/\psi\mu^+\nu_\mu)}, \tag{2}$$

with the $\tau^+$ decaying leptonically to $\mu^+\,\nu_\mu\,\bar{\nu}_\tau$. In this analysis, a sample of pp collision data corresponding to 3fb$^{-1}$ was collected with center-of-mass energies $\sqrt{s} = 7$ TeV and 8 TeV [47]. The $\tau$ lepton was reconstructed, and the global fit was performed on the missing mass squared, the $q^2$, and the decay time of $B_c$. In all these cases, the decay branching fractions are found constantly deviating from SM predictions which are currently in the range of 0.25–0.28 [48–50] which is about 2 $\sigma$ lower, see Table 1. The spread of SM predictions is due to different modeling approaches for determining the form factors [49,51]. This anomaly between observed data and the SM predictions hints at the violation of LFU. As the *B*-factories operate on the Y(4*S*) resonance for a majority of their data taking, measurements using other $B_q$ species are possible at the LHC. Furthermore, tree-level LFU tests are ongoing at LHCb, including $R(D^+)$ and the baryonic observables $R_{\Lambda_c^*}$.

There has been an accumulation of anomalies in LFU measurements in $b \to sl^+l^-$ transitions [10,14,21,34,37,38,52] which is also a fertile ground for extracting new physics (NP) signals showing a coherent pattern of deviations from the SM predictions with a

significance of 3.1 $\sigma$ and a combined statistical and systematic uncertainty of around 5 % for the LFUV observables:

$$R_{K^{(*)}} = \frac{Br(B \to K^{(*)}\mu^+\mu^-)}{Br(B \to K^{(*)}e^+e^-)} \,, \quad R_\phi = \frac{Br(B_s \to \phi\mu^+\mu^-)}{Br(B_s \to \phi e^+e^-)} \,, \tag{3}$$

where $R_{K^{(*)}}$ represents $K$ and $K^*$. These observables are predicted in the SM to be unity with uncertainties below 1% [53,54]. Here, the momentum transfer to the lepton pair is sufficiently large. Recently, LHCb confirmed these transitions to be free from anomalies and were clean observables. LHCb tested the muon–electron universality using $B^+ \to Kl^+l^-$ and $B^0 \to K^0l^+l^-$ decays. The analysis used the data of $B$ mesons from Run 1 and Run 2 corresponding to an integrated luminosity of 9 fb$^{-1}$ and announced these measurements to be in agreement with SM predictions [55]. The measurements of $R_K$ and $R_{K^*}$ for the dilepton invariant-mass squared, $q^2$, intervals $0.1 < q^2 < 1.1$ GeV$^2/c^4$ and $1.1 < q^2 < 6.0$ GeV$^2/c^4$, reported here supersede the previous LHCb measurements [10] and are compatible with the SM values. We revisit these transitions theoretically later in the construction of these observables and their degree of tension with the SM. The summary of experimental measurements of world averages with the SM predictions has been reported in Table 1.

**Table 1.** Summary of experimental measurements with the SM predictions

| LFU Parameters | LHCb Measurements | SM Prediction | Deviation |
|:---:|:---:|:---:|:---:|
| $R_D^{LHCb2022}$ | $0.441 \pm 0.060 \pm 0.066$ [56] | $0.298 \pm 0.004$ [43] | $2.16\,\sigma$ |
| $R_{D^*}^{LHCb2022}$ | $0.281 \pm 0.018 \pm 0.024$ [56] | $0.254 \pm 0.005$ [43] | $2.26\,\sigma$ |
| $R_{J/\psi}$ | $0.71 \pm 0.17 \pm 0.181$ [47] | $0.283 \pm 0.048$ [50] | $2\,\sigma$ |

Meanwhile, Quantum Chromodynamics (QCD)—the theory of strong interaction, is expected to address the problem of quark confinement inside the hadron, which in principle, is governed by the internal dynamics of constituent quarks and gluons. The decay transition rates are written in terms of Lorentz invariant form factors that encode information about hadrons as bound state systems. However, it has not been possible to extract these form factors from a straightforward first principle of the QCD hypothesis because of the complexities arising from non-abelian and non-perturbative characteristics of QCD. Therefore, alternate routes in the form of phenomenological models are considered to find inroads in describing the bound state nature of hadrons and their decay properties. The study of LFUV can be undertaken theoretically through various model-dependent as well as model-independent approaches. In literature, there are path-breaking theoretical approaches explaining the anomalies of $B$-decays involving an extension to SM couplings:

- The use of heavy quark effective theory (HQET) in parametrizing the form factors and generating order-by-order relations in $1/m_Q$ and $\alpha_s$.
- Various quark models and other potential models that approximately compute the form factors (in various kinematic regimes of $q^2$), such as the QCD sum rule, light cone sum rule, Bethe Salpeter approach, and relativistic quark model approaches.
- There are also theoretical calculations based on Lattice QCD (LQCD), which are presently available only for a limited subset of form factors and kinematic regimes. The beauty of all these theoretical developments is that they allow model-independent predictions on hadronic phenomena and test the electroweak theory in the SM.

Another prime candidate to explain the current intriguing hints for an LFUV is the vector leptoquark SU(2) singlet, see for instance Refs. [57–59]. In these works, a phenomenological analysis was done, and loop effects inside the model were calculated and studied in order to explain $R(D)$ and $R(D^*)$ involving extra pairs of fermions in the SU(4) representation modifying the original Pati–Salam (PS) model [60,61]. Authors of ref. [62] discussed these



decay transitions, $b \rightarrow s\mu^+\mu^-$, $B \rightarrow K^{(*)}\nu\bar{\nu}$, $B \rightarrow D^{(*)}\tau\nu$, $B \rightarrow K^{(*)}\tau\mu$ using gauge-invariant dim-6 operators to study these world averages. In this approach, the authors concluded that the couplings of NP are with the third generation of quarks and lepton in their interaction eigen basis. They considered the data of $R(D^*)$ and simultaneously also explained $R_K$.

In the brief outline of various models as described above, we could not be certainly exhaustive in our references. Nevertheless, it can be noted that all such models, be they non-relativistic, relativistic, QCD-inspired, or purely phenomenological, have their own advantages as well as limitations. A quark potential model description is successful if it can reproduce more or less the available observed data in various hadron sectors. No matter what is the Lorentz structure of the interaction potential used, the phenomenological model framework is considered reliable as long as it can provide a description of constituent-level dynamics inside the hadron core and predict various hadronic properties, including their decays. However, the process of parameterization at the potential level always involves a fair degree of arbitrariness. In that sense, the potential model approach is not unique, particularly when one sticks to reproducing the experimental data in a limited range only. Therefore, it is necessary to stretch the applicability of a quark model to a wider range of observed data.

In this context, we present our results on anomalies of *B*-decays in a potential model-dependent framework, i.e., a Relativistic Independent Quark Model (RIQM), which we have briefly discussed in the latter part of the article. In this paper, recent results in *B*-hadron decays are presented with a focus on testing the applicability of the RIQM framework in explaining the LFU ratios in addition to the prediction from model-independent studies as well. Section 2 consists of an experimental outlook on LFU. Sections 3 and 4 discuss the model-dependent and independent studies of $b \rightarrow c$ and $b \rightarrow s$ decays, and their corresponding results have been reported. Finally, we present our conclusions and future outlook in Section 5. For the sake of completeness, the details of our model, such as the quark orbitals, momentum probability amplitude, and the parametrization of weak decay form factors, have been mentioned in Appendices A and B.

## 2. Experimental Outlook on Lepton Flavor Universality

Since the discovery of the *b* quark in 1977 [63], large samples of *B*-hadrons have been produced at colliders such as CESR, LEP, or Tevatron. However, until the advent of the *B*-factories and the LHC, even with their specialized detectors and larger samples, it was not feasible to study third-generation LFUV in *B* mesons. *B* meson decay measurements are divided into two categories. One includes decay, which is FCNC and involves a transition from a *b*-quark to a *s*-quark with the emission of lepton pairs. These decays are heavily suppressed at the tree level due to phase space effects and can only happen at the higher order, as shown in Figure 1a. Therefore, to understand the dynamics of such decay processes, the NP mediators such as leptoquarks [64–67] and $Z'$ [68–71] should modify their amplitudes significantly. The corresponding Feynman diagram involving leptoquark is shown in Figure 1b. The second category includes the decays involving FCCC, i.e., from a *b*-quark to a *c*-quark with the emission of leptons and neutrinos, as shown in Figure 2. These decays happen at the tree level and thus have a large *Br* (up to a few percent) than $b \rightarrow sl^+l^-$ decays (Br$\sim10^{-6}$–$10^{-7}$). Nevertheless, these decays are experimentally challenging due to the presence of neutrinos in the final state. Unfolding the true nature of neutrinos may pave a clear way to a unified theory of physics.

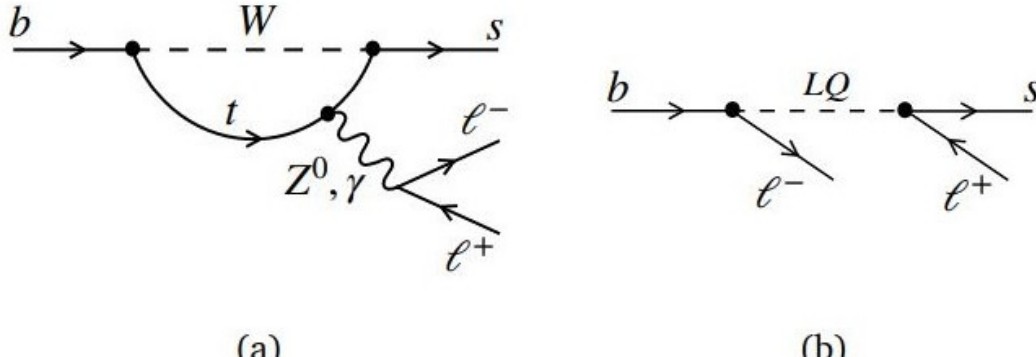

**Figure 1.** (**a**) SM contribution for $b \to s l^+ l^-$ transition including W, Z, and $\gamma$ via loop level. (**b**) NP contribution involving a leptoquark (LQ coupling directly to quarks and leptons) [72].

The LHCb detector started taking data in 2010 and has recorded unprecedented trillions of pairs of $b\bar{b}$ as of 2020, which allows it to compensate for a more challenging environment of pp collisions [73]. At hadron colliders such as the LHCb, $b$ quarks are predominantly pair-produced in pp collisions via the gluon fusion process $gg \to b\bar{b}$ with an approximate production cross-section $\sigma(b\bar{b}) \sim 560 \ \mu$b at $\sqrt{s} = 13$ TeV [5]. LHCb detector is mainly used for the study of heavy meson decays. The $b\bar{b}$ production at the LHC is mainly induced by gluon–gluon fusion where the production of two $b$-quarks is collinear and close to the beam directions. This attribute primarily influenced the design of the LHCb detector. The detector has good particle identification performances from the two RICH detectors, the electromagnetic calorimeter and the muon station. It also has excellent momentum resolution ($\Delta p / p = 0.5\%$ at low momentum) [74]. The most obvious effect of LFU tests in LHCb is the difference in the efficiencies of the electron and muon hardware triggers for the kinematics of interest. While running at the $\Upsilon(4S)$ resonances, all the $B$-hadron species (e.g., $B^+, B^0, B_s, B_c, \Lambda_b$, etc.) are produced at the LHC, and LFU tests can be performed utilizing all types of hadrons. LHCb also studies the decays of $B_c$ mesons, in spite of its very low production rate, approximately 0.6% of the $B^+$ production cross-section [75]. The recent announcement of neutral-current anomalies to be the clean ratios supersede the previous LHCb measurements. The results of $R_K$ and $R_{K^*}$ differing from previous measurements are partly due to the use of tighter electron identification criteria and partly due to the modeling of the residual misidentified hadronic backgrounds; statistical fluctuations make a smaller contribution to the difference since the same data are used as in ref. [10]. The systematic uncertainties associated with these measurements remain significantly smaller than the statistical uncertainties, which are expected to reduce further with more data collection in Run 3.

At the same time, the Belle detector is also dedicated to heavy meson decays. It is made up of a large superconducting solenoid coil that provides a 1.5 T magnetic field. Inside it, there are various components, including a silicon vertex detector, a 50-layer central drift chamber, an array of aerogel threshold Cherenkov Counters, a barrel-like arrangement of time-of-flight scintillation counter, and an electromagnetic calorimeter comprised of Si(Ti) crystals. In various particle colliders, the fine segment of strip detectors plays a vital role in understanding the beam–beam dynamics and the decay vertices of long-lived particles. With the aid of silicon strip detectors and several layers of gaseous detectors, the momentum measurements of charged particles and their trajectories become straight-forward. $B \to D(D^*) l \nu_l$ decays have already been studied in BaBar [30,31], Belle [32,33,76] and LHCb [36,77]. The upgraded Belle detector, Belle II [11], started taking data in 2018 with the aim of recording a total of over 40 billion $B\bar{B}$ pairs. The LFUV prospects for Belle II are discussed briefly in [5].

The recently assigned Belle II experiment and the LHCb detector to be upgraded in 2019–21 and 2031, respectively, are expected to continue taking data over the next decade and a half, outshining the current data samples by more than one order of magnitude.

Measurements from the newly started Belle II run from 2020 [78,79] are also expected to shed light on the current flavor anomalies with the added reliability of a complementary experimental setup. For example, the LHCb uncertainty on the $R_{D*}$ ratio is expected to scale down about a factor of 2 with the LHC Run 3, and Belle II will have enough data by then to provide an $R_D$ measurement with uncertainty 2 to 3 times smaller than the current world average [80] aligning with the SM predictions.

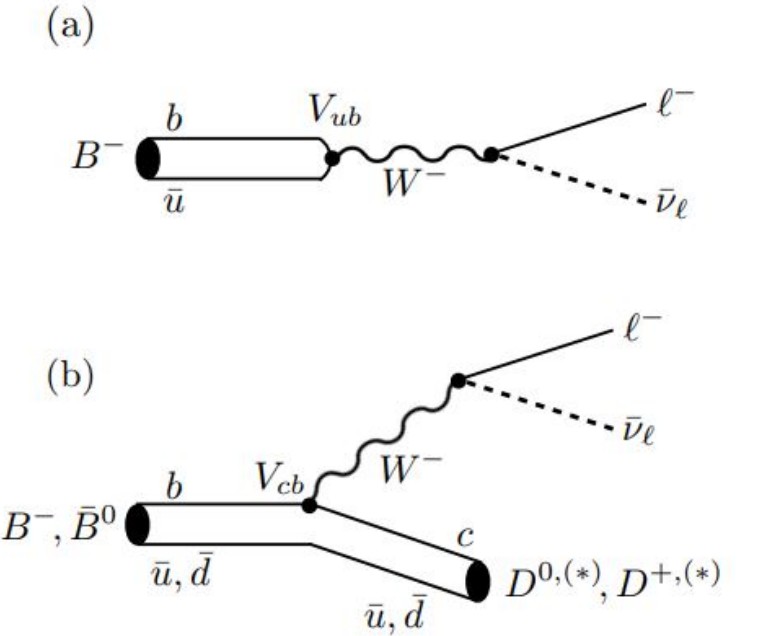

**Figure 2.** Tree -level decay processes: (**a**) $B^- \to l^- \bar{\nu}_l$ [81], (**b**) $B^- \to D^* l^- \bar{\nu}_l$ [81].

### 3. Tests of LFU Violation in $b \to c l \nu_l$ Transitions

In this section, we introduce the ratios of $b \to \tau, \mu$ leptons. Because of their significant mass difference, the decay amplitudes are believed to differ from one another. We focus on the SM tree-level description of $B_c \to X l \nu_l$, where $X = \eta_c, J/\psi, D(D^*)$. The corresponding Feynman diagram is shown in Figure 3. The study of $B_c$ meson provides an interesting area because of its unique characteristics. It is the lowest bound state of two heavy open flavored quarks (charm and bottom). The mesons in the $B_c$ family lie intermediate in mass and size between the charmonium $(c\bar{c})$ and bottomonium $(b\bar{b})$ families, where the heavy quark interactions are believed to be understood rather well. However, $B_c$-meson with two explicit heavy flavors has not yet been thoroughly studied because of insufficient data available in this sector. The $c\bar{c}$ and $b\bar{b}$ systems with hidden flavors decay via strong and electromagnetic interactions, whereas a $B_c$ meson with open flavors decays only via the weak interaction since it lies below the $B\bar{D}$ threshold. Therefore, it has a comparatively long lifetime and very rich weak decay channels with sizable branching ratios. Recently, CMS Collaboration [82] has detected excited $B_c$ state through the study of $B_c^+ \pi^+ \pi^-$ based on the entire LHC sample of pp collisions by using a total integrated luminosity of 143 fb$^{-1}$ at $\sqrt{s} = 13$ TeV which yielded $B_c(2S)$ meson mass, $6871 \pm 1.2 \pm 0.8$ MeV. It has not yet been possible to detect the ground and excited state of $B_c^*$. Hopefully, with the available energy and higher luminosity at LHC and at $Z_0$ factory, the event accumulation rate for these undetected states can be enhanced in the near future, providing scope for detailed studies of $B_c$ and $B_c^*$ counterparts. The recently observed data and the possibility of high statistics $B_c$ events expected in the ongoing and upcoming experiments provide the necessary motivation to investigate various decay properties in this sector. Thus, $B_c$-meson provides a unique window into heavy quark dynamics and gives scope for an independent test of QCD.

With this contention in mind, we present a model-dependent as well as a model-independent discussion to accommodate the wide discrepancies between the SM and BSM physics.

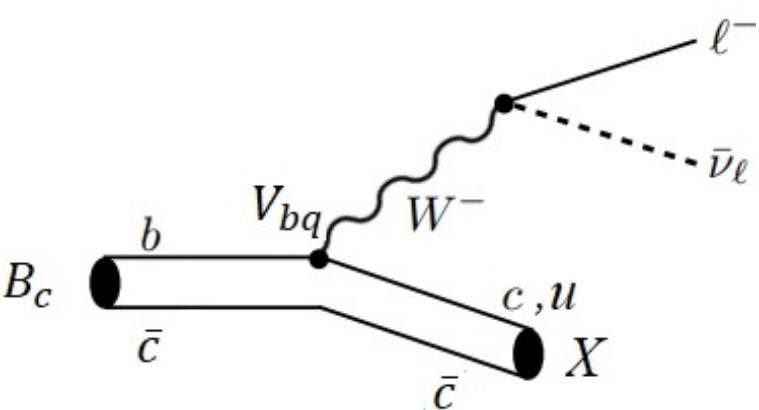

**Figure 3.** SM contribution for $B_c \to Xl\nu_l$ where $X = \eta_c, J/\psi, D(D^*)$ [81].

### 3.1. Model-Dependent Studies

The study of exclusive semileptonic decays involving the non-perturbative hadronic matrix elements is non-trivial. For reliable measurements of the invariant transition amplitudes, the rigorous field theoretic techniques, and formulation from the first principle of QCD application have not yet been possible. Therefore, various theoretical approaches employ phenomenological models to probe the non-perturbative QCD dynamics. Several theoretical approaches [48,49,83,84] exist to parameterize $B \to D(D^*)$ transitions. The use of HQET in generating the relation in $1/m_Q$ and $\alpha_s$ between form factors has been useful. In the literature, there is a plethora of quark models that approximately calculate the form factors, such as the QCD Sum rule, light cone sum rule approaches, and lattice QCD calculation. The details of these approaches to the form factors parametrization facilitate to shape of the differential decay amplitudes and provide sensitive measurements on the newfound scale in physics. Based on this, we have also presented here a short overview of the results of the world averages in the framework of a model-dependent approach. The model we adopted here is the RIQM. It is based on confining harmonic potential in the equally mixed scalar–vector form [81]

$$U(r) = \frac{1}{2}\left(1 + \gamma^0\right) V(r),$$ (4)

where $V(r) = (ar^2 + V_0)$. Here $r$ is the relative distance between quark and antiquark, $\gamma^0$ is the time-like Hermitian matrix, whereas $a$ and $V_0$ are the potential parameters that have been fixed from the earlier level of the model application using hadron spectroscopy whose values are [81]

$$(a, V_0) = \left(0.017166 \, \text{GeV}^3, -0.1375 \, \text{GeV}\right).$$

The internal dynamics of the constituent quarks are presumed to be represented through a quark Lagrangian density with a suitable Lorentz structure in the form

$$\mathcal{L}_q^0(x) = \bar{\psi}_q(x) \left[\frac{i}{2}\gamma^\mu \partial_\mu - m_q - U(r)\right] \psi_q(x).$$ (5)

This leads to the Dirac equation for individual quark as

$$\left[\gamma^0 E_q - \vec{\gamma}.\vec{p} - m_q - U(r)\right]\psi_q(\vec{r}) = 0,$$ (6)

where $\psi_q(\vec{r})$ represents the four-component Dirac normalized wave function which can be written in a two-component form as

$$\psi_q(\vec{r}) = \begin{pmatrix} \psi_A(\vec{r}) \\ \psi_B(\vec{r}) \end{pmatrix} = \begin{pmatrix} \psi_A(\vec{r}) \\ \left( \dfrac{\vec{\sigma} \cdot \vec{p}}{E_q + m_q} \right) \psi_A(\vec{r}) \end{pmatrix}. \tag{7}$$

Here, $\psi_A(\vec{r})$ and $\psi_B(\vec{r})$ are the upper and lower components of quark wave functions with opposite parity, respectively.

Like all other potential models, RIQM is a QCD-inspired phenomenological model that specifies the nature of the confinement of constituent quarks inside a hadron through an interaction potential of a suitable Lorentz structure. All the observable properties of composite hadrons are supposed to be theoretically derivable from their constituent level dynamics in terms of the bound quark eigenmodes, which have not yet been possible from the first principle of QCD due to complications arising inherently in QCD. Therefore, phenomenological routes have been sought by forcing a description of the quark dynamics through several effective quark potential models cited in the literature. The potential taken in the form is assumed to represent the non-perturbative multi-gluon interaction, whereas the residual interactions due to quark–pion coupling arising out of the restoration of chiral symmetry in PCAC limit in the SU(2)-sector and that due to one gluon exchange at a short distance are treated perturbatively in this model. The choice of the confining potential $U(r)$, reported in Equation (4), of such Lorentz structure, results in simple and tractable form when analyzing various hadronic properties and adequate tree-level description for decays involving FCCC. With this objective, we extend the applicability of the RIQM framework further to different sectors such as to show that this model provides one of the suitable alternative phenomenological schemes to study various hadronic phenomena and compare our results with that of other theoretical attempts and the available experimental data. The applicability and accountability of this model have already been tested in describing a wide range of hadronic phenomena, including the radiative, weak radiative, rare radiative [85–90], leptonic [91], weak leptonic [92], semileptonic [93–95], radiative leptonic [96–98], and non-leptonic [99–102] decays of hadrons in the light and heavy flavor sectors [72,81,103]. The invariant matrix element for $B_c \to \eta_c (J/\psi) l^- \bar{\nu}_l$ and $B_c \to D(D^*) l^- \bar{\nu}_l$ is written in the general form as [104]

$$\mathcal{M}(p, k, k_l, k_\nu) = \frac{\mathcal{G}_\mathcal{F}}{\sqrt{2}} V_{bq'} \, \mathcal{H}_\mu(p, k) \, \mathcal{L}^\mu(k_l, k_\nu), \tag{8}$$

where $\mathcal{G}_F$ is the effective Fermi coupling constant, $V_{bq'}$ is the relevant CKM parameter, and $\mathcal{L}^\mu$ and $\mathcal{H}_\mu$ are leptonic and hadronic current, respectively. Here, $p, k, k_l,$ and $k_\nu$ denote parent ($B_c$) and daughter (X) mesons, a lepton, and the neutrino four-momentum, respectively. The decay process physically takes place when participating mesons are in their momentum eigenstates. Therefore, in the field-theoretic description of any decay process, it is necessary to represent the meson-bound states by appropriate momentum wave packets reflecting momentum and spin distribution between constituent quark and antiquark inside the meson core. The details of the model have been discussed in Appendices A and B. In the RIQM approach, the wave packet representing a meson bound state, for example, $|B_c(\vec{p}, S_{B_c})\rangle$, at a definite momentum $\vec{p}$ and spin $S_{B_c}$ takes the form [66–71,74]

$$\left| B_c(\vec{p}, S_{B_c}) \right\rangle = \hat{\Lambda}(\vec{p}, S_{B_c}) \left| (\vec{p}_b, \lambda_b); (\vec{p}_c, \lambda_c) \right\rangle, \tag{9}$$

where $\left| (\vec{p}_b, \lambda_b); (\vec{p}_c, \lambda_c) \right\rangle$ is the Fock space representation of the unbound quark and antiquark in a color–singlet configuration with their respective momentum and spin, while $\hat{\Lambda}(\vec{p}, S_{B_c})$ represents an integral operator which encodes the bound state characteristic of a meson.

Incorporating the weak form factors, which are derived using the covariant expansion of hadronic amplitudes from the model dynamics, the angular decay distribution in the momentum transfer squared $q^2$, ( $q = p - k = k_l + k_v$ ) is obtained as [104]

$$\frac{d\Gamma}{dq^2 d\cos\theta} = \frac{\mathcal{G}_F}{(2\pi)^3}|V_{bq}|^2\frac{(q^2 - m_l^2)^2}{8M^2q^2}|\vec{k}|\mathcal{L}^{\mu\sigma}\mathcal{H}_{\mu\sigma}, \tag{10}$$

where $\mathcal{L}^{\mu\sigma}$ and $\mathcal{H}_{\mu\sigma}$ are the lepton and hadron correlation functions, respectively, $m_l$ is the mass of charged lepton, and $M$ is the mass of parent ($B_c$) meson. Using the completeness property, the lepton and hadron tensors in Equation (10) can be rewritten as follows.

$$\begin{aligned}
\mathcal{L}^{\mu\sigma}\mathcal{H}_{\mu\sigma} &= \mathcal{L}_{\mu'\sigma'}g^{\mu'\mu}g^{\sigma'\sigma}\mathcal{H}_{\mu\sigma}, \\
&= \mathcal{L}_{\mu'\sigma'}\epsilon^{\mu'}(m)\epsilon^{\mu^\dagger}(m')g_{mm'}\epsilon^{\sigma^\dagger}(n)\epsilon^{\sigma'}(n')g_{nn'}\mathcal{H}_{\mu\sigma}, \\
&= L(m,n)g_{mm'}g_{nn'}H(m'n').
\end{aligned} \tag{11}$$

Here, lepton and hadron tensors are introduced in the space of helicity components:

$$\begin{aligned}
L(m,n) &= \epsilon^\mu(m)\epsilon^{\sigma^\dagger}(n)\mathcal{L}_{\mu\nu}, \\
H(m,n) &= \epsilon^{\mu^\dagger}(m)\epsilon^\sigma(n)\mathcal{H}_{\mu\nu}.
\end{aligned} \tag{12}$$

It is convenient to express physical observables on a helicity basis for the sake of simplicity. On this basis, the helicity form factors are expressed in terms of the Lorentz invariant form factors that represent the decay amplitudes. Then one can perform the Lorentz contraction in the above Equation (10) with the helicity amplitudes as done in [104]. In this analysis, we do not consider the azimuthal $\chi$ distribution of the lepton pair, and therefore, we integrate over the azimuthal angle dependence of the lepton tensor that yields the differential partial helicity rates $(d\Gamma_i/dq^2)$ in the form:

$$\frac{d\Gamma_i}{dq^2} = \frac{\mathcal{G}_f^2}{(2\pi)^3}|V_{bq'}|^2\frac{(q^2 - m_l^2)^2}{12M^2q^2}|\vec{k}|H_i, \tag{13}$$

with $H_i(i = U, L, P, S, SL)$ representing a standard set of helicity structure functions given by linear combinations of helicity components of hadron tensor $H(m,n) = H_m H_n^\dagger$ as

$$\begin{aligned}
H_U &= Re(H_+ H_+^\dagger) + Re(H_- H_-^\dagger) & &: Unpolarized - transversed, \\
H_L &= Re(H_0 H_0^\dagger) & &: Longitudinal, \\
H_P &= Re(H_+ H_+^\dagger) - Re(H_- H_-^\dagger) & &: Parity - odd, \\
H_S &= 3Re(H_t H_t^\dagger) & &: Scalar, \\
H_{SL} &= Re(H_t H_0^\dagger) & &: Scalar - Longitudinal\ Interference.
\end{aligned}$$

Our goal is to study the LFU ratio in the RIQM framework. Therefore, we evaluate the observable $R$ within the model [104]. Our predicted observable $R$ for $B_c \to X(nS)l\nu_l$, ($X = \eta_c, J/\psi, D(D^*)$) in the ground and radially excited states are found comparable to other SM predictions, as given in Tables 2 and 3, respectively. The deviation of SM predictions of $R$ from the experimental data clearly indicates anomalies in semileptonic decays and the failure of encoding NP bounds in our RIQM. However, in the absence of predicted data from established model approaches, in the literature, our predictions for LFUV observables for the charm and charmonium higher states, $R_D(2S)$, $R_D(3S)$, $R_{D^*}(2S)$, and $R_{D^*}(3S)$ can also be useful for identifying the $B_c$ channels in the upcoming Run 3 data at LHCb.

**Table 2.** Results of ratios of branching fractions for Semileptonic $B_c$-decays in the ground state.

| Ratio of Branching Fractions ($R$) | RIQM | [105] | [51] | [106] |
|---|---|---|---|---|
| $R_{\eta_c} = \frac{\mathcal{B}(B_c \to \eta_c l\nu)}{\mathcal{B}(B_c \to \eta_c \tau\nu)}$ | 2.312 | 3.96 | 3.68 | 3.2 |
| $R_{J/\psi} = \frac{\mathcal{B}(B_c \to J/\psi l\nu)}{\mathcal{B}(B_c \to J/\psi \tau\nu)}$ | 4.785 | 4.18 | 4.22 | 3.4 |
| $R_D = \frac{\mathcal{B}(B_c \to Dl\nu)}{\mathcal{B}(B_c \to D\tau\nu)}$ | 1.275 | 1.57 | 1.67 | 1.42 |
| $R_{D^*} = \frac{\mathcal{B}(B_c \to D^* l\nu)}{\mathcal{B}(B_c \to D^* \tau\nu)}$ | 1.091 | 1.76 | 1.72 | 1.66 |

**Table 3.** LFU observables for $B_c$-decays to radially excited charmonium states.

| Ratio | RIQM | [107] | [108,109] | [110] | [111] | [112] |
|---|---|---|---|---|---|---|
| $R_{\eta_c}(2S)$ | 7.33 | 18.4 | - | 14.5 | 1.35 | 35.38 |
| $R_{\eta_c}(3S)$ | 46.67 | 96.24 | $1.1 \times 10^3$ | $7.36 \times 10^2$ | 33.33 | |
| $R_\psi(2S)$ | 11.76 | 1.98 | - | 14.3 | - | 14 |
| $R_\psi(3S)$ | 12.876 | 109 | 158.33 | 947.4 | | |

It is worthwhile to note here that in our RIQM approach, the parametrization of relevant form factors of semileptonic decay amplitudes are evaluated in the entire kinematic range ($0 \leq q^2 \leq q^2_{max}$), which makes our prediction more reliable and accurate than other theoretical approaches. In other theoretical models, cited above, the form factors are first calculated with an endpoint normalization at either $q^2 = 0$ (maximum recoil point) or $q^2 = q^2_{max}$ (minimum recoil point). Then using monopoles, dipoles, and Gaussian ansatz they are phenomenologically extrapolated to the whole physical region, making form factor estimation less reliable. To dodge such uncertainties in the calculation, we do not resort to any such phenomenological ansatz instead. Given the high statistics, $B_c$-events which are expected to yield up to $10^{10}$ events in each upcoming year at the colliders, semileptonic $B_c$ decays to charm and charmonium states present a fascinating sphere to explore more and more on the new-found scale in physics.

*3.2. Model-Independent Studies*

In order to probe the nature of BSM physics, the semileptonic decays can also be executed through model-independent studies. Assuming the neutrino to be left chiral, the effective Hamiltonian for $b \to cl\nu_l$ transition containing all possible Lorentz structures is given as:

$$\mathcal{H}_{\text{eff}} = \sqrt{8}\, G_F V_{cb} \left[ \mathcal{O}_{VL} + \frac{1}{\sqrt{8} G_F V_{cb} \Lambda^2} \left( C_i O_i + C_i' O_i' + C_i'' O_i'' \right) \right], \qquad (14)$$

where $V_{cb}$ is the CKM matrix element, $G_F$ is the Fermi coupling constant, and $\mathcal{O}_{VL}$ is the SM operator, which has the usual (V-A)⊗(V-A) structure. The couplings $C_i$, $C_i'$, and $C_i''$ represents the Wilson coefficient of the NP operator in which NP effects are encoded. Authors of ref. [113] have defined $(\sqrt{8} G_F V_{cb} \Lambda^2)^{-1} \approx \alpha$, where $\Lambda$ scale is set to 1 TeV for the NP effect. This leads to $\alpha = 0.749$. The primed and double primed operators are products of a quark–lepton bilinear, and they arise from various leptoquarks models [114–117]. The authors have performed the $\chi^2$ fitting considering either one NP operator or a combination of two similar operators. Using the best-fit values they have provided the LFU parameter given in Table 4.

It is observed that the effect of various NP contributions to LFU parameters $R_\eta$ is almost negligible while values for $\mathcal{C}_{\mathcal{S}_\mathcal{L}}''$ give a marginal deviation from the SM prediction. Here, the NP coefficients $\mathcal{C}_{\mathcal{V}_\mathcal{L}}$, $\mathcal{C}_{\mathcal{V}_\mathcal{L}}'$, and $\mathcal{C}_{\mathcal{S}_\mathcal{L}}''$ are expressed in linear combination form of couplings $C_i$, $C_i'$, and $C_i''$, as done in [113]. Moreover, the vector couplings have shown

larger deviation as compared to other decay widths. Therefore, the sensitivity of new couplings on angular observable is quite a welcoming aspect for manifesting NP.

**Table 4.** Results of branching ratio and angular observables of $B_c \to \eta_c \tau^+ \nu_\tau$ and $B \to D_0^* \tau^+ \nu_\tau$ using the new complex Wilson Coefficients.

| Angular Observables | Values for SM | Values for $\mathcal{C}_{\mathcal{V}_\mathcal{L}}$ | Values for $\mathcal{C}'_{\mathcal{V}_\mathcal{L}}$ | Values for $\mathcal{C}''_{\mathcal{S}_\mathcal{L}}$ |
|---|---|---|---|---|
| $\mathrm{Br}(B_c \to \eta_c \tau^+ \nu_\tau)$ | $0.0020 \pm 0.124$ | $0.0026 \pm 0.112$ | $0.0026 \pm 0.112$ | $0.0028 \pm 0.101$ |
| $\mathcal{R}_\eta$ | $0.284 \pm 0.02$ | $0.284 \pm 0.01$ | $0.284 \pm 0.01$ | $0.353 \pm 0.06$ |
| $\mathrm{Br}(B \to D_0^* \tau^+ \nu_\tau)$ | $0.0027 \pm 0.02$ | $0.0119 \pm 0.01$ | $0.0119 \pm 0.01$ | - |

## 4. Tests of LFU Violation in Transitions

In the SM, transitions between different quark flavors can only be mediated by the charged weak bosons $W^\pm$. As a consequence, FCNC transitions between same charge quarks are not directly mediated by the neutral weak boson $Z^0$ but rather occur through much rarer loop processes involving virtual $W^\pm$ and additional virtual quarks, in penguin- and box-like Feynman diagrams. The SM predicts the dynamics of decays governed by FCNC transitions with very high precision. New particles can either participate in the loops or generate additional tree-level diagrams. The amplitudes of suppressed decays governed by $b \to s l^+ l^-$ transitions are ideal laboratories to look for NP as effects beyond the SM can be sizable with respect to the competing SM processes. Recently, there has been an accumulation of LFU measurements in these transitions showing deviation from SM predictions. Authors of ref. [10] reported evidence of LFUV with a significance of 3.1 $\sigma$ and a combined statistical and systematic uncertainty of around 5%. Decays induced at the quark level, such as $b \to s l^+ l^-$ have been given much attention due to the significant violation of lepton universality. For the first time, the LHCb collaboration reported these violation measurements [10,14,52]

$$R_K = 0.745(\mathrm{stat}) \pm 0.036(\mathrm{syst}) \,, \tag{15}$$

which deviated from the SM prediction of $R_K = 1.0004$ [53] with a significance of 2.6 $\sigma$. However, recently LHCb confirmed $R_K$ and $R_{K^*}$ measurements to be free from anomalies and compatible with SM predictions. The measured values of the interest are [55]

$$R_K = 0.994(\mathrm{stat}) \pm 0.029(\mathrm{syst}) \,, \quad R_{K^*} = 0.927(\mathrm{stat}) \pm 0.036(\mathrm{syst}) \,, \tag{16}$$

$$R_K = 0.949(\mathrm{stat}) \pm 0.022(\mathrm{syst}) \,, \quad R_{K^*} = 1.027(\mathrm{stat}) \pm 0.027(\mathrm{syst}) \,. \tag{17}$$

The measured values of $R_K$ and $R_{K^*}$ for the $q^2$ intervals: $0.1 < q^2 < 1.1$ GeV$^2$/$c^4$ corresponds to low $q^2$ region and $1.1 < q^2 < 6.0$ GeV$^2$/$c^4$ corresponds to central $q^2$ as reported here, supersede previous LHCb measurements [10] and are in agreement with the predictions of the SM. All pp collisions data recorded using the LHCb detector between 2011 and 2018 are used, corresponding to integrated luminosities of 1.0, 2.0, and 6.0 $fb^{-1}$ at center-of-mass energies of 7, 8, and 13 TeV. The systematic uncertainties associated with these measurements remain significantly smaller than the statistical uncertainties and are expected to reduce further with more data.

In the SM, these decays do not have hadronic uncertainties and can be predicted precisely due to the insignificant mass difference of electron–muon. Therefore, these decays are not allowed in the first-order process and can only participate in loop order in the SM. The suppression of these transitions can only be understood in terms of the fundamental symmetries of the SM.

In order to scrutinize the universality effects, theoretical frameworks have been used extensively; for example, the authors of Ref. [54] have used detailed analysis incorporating QED-radiative corrections in the Monte Carlo framework. They have cleanly canceled

out the hadronic uncertainties pertaining to the non-perturbative effects of QCD. In their work, they have wisely used the analytic results of meson effective theory and have found $R_K$ to be a "safe observable", i.e., an unambiguous prediction of the SM. It is interesting to scrutinize the size of these corrections from the theory side in order to identify the most sensitive moments and give further motivation to an experimental investigation. Since lepton universality violation observables also depend on the charm loop through the interference between NP and SM contributions, therefore, to obtain an unbiased picture of NP, authors of ref. [118] have used charming penguins and have solved the charm loop amplitudes while investigating unbiased NP solutions. Additional details can also be found in ref. [119] with detailed numerical comparison for neutral mode $\bar{B}^0 \to \bar{K}^0 l^+ l^-$ which is also relevant in the study of LFU ratios. The authors of ref. [120] have used the updated global fit of muon Wilson Coefficients to explain anomalies. So there is this large discrepancy of 5.6 $\sigma$ that attracts wider attention to propose various NP models.

In [38], the LHCb collaboration presented new measurements of $R_{K_S^0}$ and $R_{K^{*+}}$ and also provided updated measurements for several $B_s \to \phi \mu^+ \mu^-$ observables [39,40], which deviates from the SM by 3.6 $\sigma$ level. Therefore, a theoretical analysis of these decay transitions using the language of effective field theory in a statistical approach was undertaken, which generated a good fit to the data [121]. Probing deeper into the decay transition of $b \to s l^+ l^-$ can also give a better understanding of the observed recalcitrant disparity between the amount of matter and antimatter in the universe. Therefore, there has been extensive study of CP-violating angular observable with a complex phase which would enable a unique determination of Lorentz structure of possible NP in this transition [122].

Recently, Glashow, Guadagnoli, and Lane (GGL) [123] proposed an explanation of the $R_K$ puzzle. Using an effective field theory approach, they demonstrated that an NP model could simultaneously explain both the $R_K$ and $R_{D^{(*)}}$ puzzles. Under the theoretical assumption that the NP couples predominantly to the third generation and that the scale of NP is much larger than the weak scale, then there are two types of fully gauge-invariant NP operators that contain both neutral-current and charged-current interactions. A similar explanation of a unified theory of anomalies in the framework of effective theory can also be seen in [124–126].

## 5. Conclusions & Future Outlook

Despite the SM being the most successful mathematical framework, it is still incomplete. In this paper, we have discussed the recent landscape of LFUV anomalies emerging in *B* physics from the theoretical as well as experimental points of view, which may open new vistas in the upcoming data collection and evaluation explaining BSM physics. Recently, the LFU ratios corresponding to neutral-current ($R_K$, $R_{K^*}$) are found to be in agreement with SM predictions and are said to be theoretically clean observables. We have also presented the recently updated results for $R_D$ and $R_{D^*}$, which overall have significance in the range of 3 $\sigma$ deviation. Theoretically, these transitions have been studied in a model-dependent (RIQM) framework to test its applicability for studying LFUV ratios also in addition to the study in the model-independent framework. This discrepancy has been explained with several NP models involving leptoquark and other possible models, which include $Z'$-boson [127], composite Higgs boson [128], dark matter [129], right-handed neutrinos [130], etc. If such results are further continued and confirmed, it would be unambiguous evidence of NP interpretations. Moreover, with the start of the Run 3 data-taking period (2022–2025) at the LHC, the data collections are expected to boom approximately three times larger in three years. This will certainly increase the event statistics, which will further reduce the statistical and systematic uncertainties leading to unprecedented precision for flavor measurements. The expected increase in luminosity would help in reshaping the semileptonic analyses. Additionally, the LHCb detector will undergo several staged upgrades in upcoming years in which the removal of the hardware trigger and the replacements of several sub-detectors, such as the vertex and the tracking detectors, will reduce the background coming from charged and neutral tracks and will make the electronic

(and tauonic) modes more accessible. Belle II analysis is also of fundamental importance in order to independently clarify the flavor anomalies that have been puzzling the physics community in the last decade. On the theoretical side, we should comprehensively report the results from existing models and welcome more NP models to explain these anomalies for testing on the experimental predictions. A detailed study has been reported in ref. [5] on the upcoming Run 3 data collection and analysis and has shed light on the Future Circular Hadron Collider FCC-hh at CERN that would extend the reach for direct observation of NP mediators into the multi-TeV range.

In addition, there are other anomalies that have been currently observed in recent times, rather a strong indication of NP. The anomalous magnetic moment of the muon and electron ($a_\mu$, $a_e$) observed at Fermilab and the mass of the W-boson, which have a possible combined origin with the anomalies in $B$ meson decays. The confirmation by the Fermilab (g-2) Collaboration [131] with the old BNL result has now increased the deviation from the data-driven theory prediction from the SM to about 4.2 $\sigma$. The promising point is that each of these flavor anomalies is over 3 $\sigma$. So, the chances of surviving at least one anomaly at these crucial times would lead us to a new understanding of physics. If the LFUV anomalies stay, then in no time, there will be some remarkable evidence unraveling the NP in the flavor fraternity that will trigger an intense workout for future experimentalists as well as theorists [132].

**Author Contributions:** Conceptualization, methodology, software, validation, formal analysis, investigation, resources, data curation, writing—original draft preparation, writing—review and editing, visualization, supervision, project administration, and funding acquisition: S.P. and R.S. All authors have read and agreed to the published version of the manuscript.

**Funding:** R.S. acknowledges the support of the Polish NAWA Bekker program no.: BPN/BEK/2021/1/00342 and the support of the Polish National Science Centre, Grant No. 2018/30/E/ST2/00432.

**Data Availability Statement:** No new data were created or analyzed in this study. Data sharing is not applicable to this article.

**Conflicts of Interest:** The authors declare no conflict of interest. The funders had no role in the design of the study; in the collection, analyses, or interpretation of data; in the writing of the manuscript; or in the decision to publish the results.

## Appendix A. Quark Orbitals, Momentum Probability Amplitudes, and Meson States

In the RIQM framework, a meson is defined as a color–singlet assembly of a quark and an antiquark confined independently by an effective average flavor independent potential in the form: $U(r) = \frac{1}{2}(1 + \gamma^0)(ar^2 + V_0)$ where ($a$, $V_0$) are the potential parameters. The zeroth-order quark dynamics generated by the phenomenological potential $U(r)$ taken in equally mixed scalar–vector harmonic form provide an adequate description of the transitions being analyzed in this work. Incorporating interaction potential $U(r)$ in the zeroth-order quark Lagrangian density, the ensuing Dirac equation gives a static solution of positive and negative energy quark orbitals as

$$
\begin{aligned}
\psi_\xi^{(+)}(\vec{r}) &= \begin{pmatrix} \frac{ig_\xi(r)}{r} \\ \frac{\vec{\sigma}.\hat{r}f_\xi(r)}{r} \end{pmatrix} U_\xi(\hat{r}), \\
\psi_\xi^{(-)}(\vec{r}) &= \begin{pmatrix} \frac{i(\vec{\sigma}.\hat{r})f_\xi(r)}{r} \\ \frac{g_\xi(r)}{r} \end{pmatrix} \tilde{U}_\xi(\hat{r}),
\end{aligned}
\tag{A1}
$$

respectively, where $\xi = (nlj)$ represents a set of Dirac quantum numbers specifying the eigenmodes, and $U_\xi(\hat{r})$ and $\tilde{U}_\xi(\hat{r})$ are the spin angular parts expressed, respectively, as

$$
\begin{aligned}
U_{ljm}(\hat{r}) &= \sum_{m_l, m_s} < lm_l \frac{1}{2} m_s | jm > Y_l^{m_l}(\hat{r}) \chi_{\frac{1}{2}}^{m_s}, \\
\tilde{U}_{ljm}(\hat{r}) &= (-1)^{j+m-l} U_{lj-m}(\hat{r}).
\end{aligned}
\tag{A2}
$$

The quark binding energy $E_q$ and quark mass $m_q$ are expressed as $E_q' = (E_q - V_0/2)$ and $m_q' = (m_q + V_0/2)$, respectively. One can obtain solutions to the resulting radial equation for $g_\xi(r)$ and $f_\xi(r)$ as

$$
\begin{aligned}
g_{nl} &= N_{nl}(\frac{r}{r_{nl}})^{l+l} \exp(-r^2/2r_{nl}^2) L_{n-1}^{l+1/2}(r^2/r_{nl}^2), \\
f_{nl} &= N_{nl}(\frac{r}{r_{nl}})^l \exp(-r^2/2r_{nl}^2), \\
&\times \left[ (n+l-\frac{1}{2}) L_{n-1}^{l-1/2}(r^2/r_{nl}^2) + n L_n^{l-1/2}(r^2/r_{nl}^2) \right],
\end{aligned}
\tag{A3}
$$

where $r_{nl} = a\omega_q^{-1/4}$, with $\omega_q = E_q' + m_q'$, is a state independent length parameter and $N_{nl}$ is an overall normalization constant given by

$$
N_{nl}^2 = \frac{4\Gamma(n)}{\Gamma(n+l+1/2)} \frac{(\omega_{nl}/r_{nl})}{(3E_q' + m_q')},
\tag{A4}
$$

and $L_{n-1}^{l+1/2}(r^2/r_{nl}^2)$ etc. are associated with Laguerre polynomials. The radial solutions give an independent quark bound-state condition in the form of a cubic equation

$$
\sqrt{(\omega_q/a)}(E_q' - m_q') = (4n + 2l - 1).
\tag{A5}
$$

The solution to the cubic equation provides the zeroth-order binding energies of the confined quark and antiquark for all possible eigenmodes.

In the RIQM framework, the constituent quark and antiquark are thought to move independently inside the $B_c$-meson bound state with momentum $\vec{p}_b$ and $\vec{p}_c$, respectively. Their individual momentum probability amplitudes are obtained via momentum projection of respective quark orbitals (A1) in the hadron core. In the present model, we consider the state of a meson as a wave packet representation, $|B_c(\vec{P}, S_{B_c})\rangle$, at momentum $\vec{P}$ and spin projection $S_{B_c}$ in the form as:

$$
|B_c(\vec{P}, S_{B_c})\rangle = \hat{\Lambda}_{B_c}(\vec{P}, S_{B_c}) |(\vec{p}_b, \lambda_1); (\vec{p}_c, \lambda_2)\rangle,
\tag{A6}
$$

where $\hat{\Lambda}_{B_c}(\vec{P}, S_{B_c})$ represents an integral operator

$$
\hat{\Lambda}_{B_c}(\vec{P}, S_{B_c}) = \frac{\sqrt{3}}{\sqrt{N(\vec{P})}} \sum_{\delta_1, \delta_2} \zeta_{1,2}(\lambda_1, \lambda_2) \int d^3\vec{p}_b \, d^3\vec{p}_c \, \delta^{(3)}(\vec{p}_b + \vec{p}_c - \vec{P}) \mathcal{G}_{B_c}(\vec{p}_b, \vec{p}_c).
\tag{A7}
$$

Here, $\sqrt{3}$ is the effective color factor and $\zeta_{1,2}(\lambda_1, \lambda_2)$ stands for SU(6)-spin flavor coefficients for the meson $B_c$. $N(\vec{P})$ is the meson-state normalization, which is expressed in an integral form

$$
N(\vec{P}) = \int d\vec{p}_b \, |\mathcal{G}_{B_c}(\vec{p}_b, \vec{P} - \vec{p}_b)|^2,
\tag{A8}
$$

and $\mathcal{G}_{B_c}(\vec{p}_b, \vec{p}_c)$ is the effective momentum profile function showing the distribution of momentum of individual quark and antiquark inside the meson core. In the present model description of the relativistic independent constituent quarks, we take $\mathcal{G}_{B_c}(\vec{p}_b, \vec{p}_c)$

in the form of a geometric mean of constituent quark–antiquark momentum probability amplitudes as

$$\mathcal{G}_{B_c}(\vec{p}_b, \vec{p}_c) = \sqrt{G_b(\vec{p}_b)\tilde{G}_c(\vec{p}_c)}. \tag{A9}$$

For ground-state mesons ($n = 1, l = 0$), we have

$$G_b(\vec{p}_b) = \frac{i\pi\mathcal{N}_b}{2\alpha_b\omega_b}\sqrt{\frac{(E_{p_b} + m_b)}{E_{p_b}}}(E_{p_b} + E_b) \times \exp\left(-\frac{\vec{p}_b^{\,2}}{4\alpha_b}\right),$$

$$\tilde{G}_c(\vec{p}_c) = -\frac{i\pi\mathcal{N}_c}{2\alpha_c\omega_c}\sqrt{\frac{(E_{p_c} + m_c)}{E_{p_c}}}(E_{p_c} + E_c) \times \exp\left(-\frac{\vec{p}_c^{\,2}}{4\alpha_c}\right). \tag{A10}$$

For the excited meson state ($n = 2, l = 0$), we have

$$G_b(\vec{p}_b) = \frac{i\pi\mathcal{N}_b}{2\alpha_b}\sqrt{\frac{(E_{p_b} + m_b)}{E_{p_b}}}\frac{(E_{p_b} + E_b)}{(E_b + m_b)} \times \left(\frac{\vec{p}_b^{\,2}}{2\alpha_b} - \frac{3}{2}\right)\exp\left(-\frac{\vec{p}_b^{\,2}}{4\alpha_b}\right),$$

$$\tilde{G}_c(\vec{p}_c) = \frac{i\pi\mathcal{N}_c}{2\alpha_c}\sqrt{\frac{(E_{p_c} + m_c)}{E_{p_c}}}\frac{(E_{p_c} + E_c)}{(E_c + m_c)} \times \left(\frac{\vec{p}_c^{\,2}}{2\alpha_c} - \frac{3}{2}\right)\exp\left(-\frac{\vec{p}_c^{\,2}}{4\alpha_c}\right). \tag{A11}$$

Finally, for the excited meson state ($n = 3, l = 0$), we have

$$G_b(\vec{p}_b) = \frac{i\pi\mathcal{N}_b}{2\alpha_b}\sqrt{\frac{(E_{p_b} + m_b)}{E_{p_b}}}\frac{(E_{p_b} + E_b)}{(E_b + m_b)} \times \left(\frac{\vec{p}_b^{\,4}}{8\alpha_b^2} - \frac{5\vec{p}_b^{\,2}}{4\alpha_b} + \frac{15}{8}\right)\exp\left(-\frac{\vec{p}_b^{\,2}}{4\alpha_b}\right),$$

$$\tilde{G}_c(\vec{p}_c) = \frac{i\pi\mathcal{N}_c}{2\alpha_c}\sqrt{\frac{(E_{p_c} + m_c)}{E_{p_c}}}\frac{(E_{p_c} + E_c)}{(E_c + m_c)} \times \left(\frac{\vec{p}_c^{\,4}}{8\alpha_c^2} - \frac{5\vec{p}_c^{\,2}}{4\alpha_c} + \frac{15}{8}\right)\exp\left(-\frac{\vec{p}_c^{\,2}}{4\alpha_c}\right). \tag{A12}$$

The binding energy for the constituent quark and antiquark in their ground and radially excited final meson states for $n = 1, 2, 3$; $l = 0$ can also be obtained by solving respective cubic equations representing appropriate bound state conditions.

**Appendix B. Parametrization of Weak Decay Form Factors**

Incorporating the meson states from the model dynamics, the hadronic amplitude $\mathcal{H}_\mu$ in the $B_c$-rest frame is obtained as

$$\mathcal{H}_\mu = \sqrt{\frac{4ME_k}{N_{B_c}(0)N_X(\vec{k})}} \int \frac{d^3p_b}{\sqrt{2E_{p_b}2E_{k+p_b}}}\mathcal{G}_{B_c}(\vec{p}_b, -\vec{p}_b)\mathcal{G}_X(\vec{k} + \vec{p}_b, -\vec{p}_b)\langle S_X|J_\mu^h(0)|S_{B_c}\rangle, \tag{A13}$$

where $E_{p_b}$ and $E_{k+p_b}$ stand for the energy of the non-spectator quark of the parent and daughter meson, respectively, and $\langle S_X|J_\mu^h(0)|S_{B_c}\rangle$ represents symbolically the spin matrix elements of vector–axial vector current. For the parametrization of the form factors, the hadronic amplitudes are covariantly expanded in terms of a set of Lorentz invariant form factors.

In ($0^- \to 0^-$) type transitions, it is defined as:

$$\mathcal{H}_\mu(B_c \to (\bar{c}c/\bar{u}c)_{S=0}) = (p + k)_\mu F_+(q^2) + q_\mu F_-(q^2). \tag{A14}$$

In ($0^- \to 1^-$) type transitions, the expansion is:

$$\begin{aligned}
\mathcal{H}_\mu(B_c \to (\bar{c}c/\bar{u}c)_{S=1}) = &\frac{1}{(M + m)}\epsilon^{\sigma\dagger}\Big\{g_{\mu\sigma}(p + k)qA_0(q^2), \\
&+ (p + k)_\mu(p + k)_\sigma A_+(q^2) + q_\mu(p + k)_\sigma A_-(q^2), \\
&+ i\epsilon_{\mu\sigma\alpha\beta}(p + k)^\alpha q^\beta V(q^2)\Big\}. \tag{A15}
\end{aligned}$$

The axial vector current does not contribute in $0^- \to 0^-$ transitions. The spin matrix elements corresponding to the non-vanishing vector current parts are obtained in the form

$$\langle S_X(\vec{k})|V_0|S_{B_c}(0)\rangle = \frac{(E_{p_b} + m_b)(E_{p_{c/u}} + m_{c/u}) + |\vec{p}_b|^2}{\sqrt{(E_{p_b} + m_b)(E_{p_{c/u}} + m_{c/u})}}, \tag{A16}$$

$$\langle S_X(\vec{k})|V_i|S_{B_c}(0)\rangle = \frac{(E_{p_b} + m_b)k_i}{\sqrt{(E_{p_b} + m_b)(E_{p_b+k} + m_{c/u})}}. \tag{A17}$$

From the above spin matrix elements, the expressions for hadronic amplitudes are compared yielding the form factors $f_+$ and $f_-$ for $0^- \to 0^-$ transition as

$$f_\pm(q^2) = \frac{1}{2M}\sqrt{\frac{ME_k}{N_{B_c}(0)N_X(\vec{k})}} \int d\vec{p}_b \mathcal{G}_{B_c}(\vec{p}_b, -\vec{p}_b)\mathcal{G}_X(\vec{k} + \vec{p}_b, -\vec{p}_b),$$

$$\times \frac{(E_{o_b} + m_b)(E_{p_{c/u}} + m_{c/u}) + |\vec{p}_b|^2 \pm (E_{p_b} + m_b)(M \mp E_k)}{E_{p_b} E_{p_{c/u}}(E_{p_b} + m_b)(E_{p_{c/u}} + m_{c/u})}. \tag{A18}$$

For $(0^- \to 1^-)$ transitions, the spin matrix elements corresponding to the vector and axial–vector currents are found separately in the form:

$$\langle S_X(\vec{k}, \hat{\epsilon}^*)|V_0|S_{B_c}(0)\rangle = 0, \tag{A19}$$

$$\langle S_X(\vec{k}, \hat{\epsilon}^*)|V_i|S_{B_c}(0)\rangle = \frac{i(E_{p_b} + m_b)(\hat{\epsilon}^* \times \vec{k})_i}{\sqrt{(E_{p_b} + m_b)(E_{p_b+k} + m_{c/u})}}, \tag{A20}$$

$$\langle S_X(\vec{k}, \hat{\epsilon}^*)|A_i|S_{B_c}(0)\rangle = \frac{(E_{p_b} + m_b)(E_{p_b+k} + m_{c/u}) - \frac{|\vec{p}_b|^2}{3}}{\sqrt{(E_{p_b} + m_b)(E_{p_b+k} + m_{c/u})}}, \tag{A21}$$

$$\langle S_X(\vec{k}, \hat{\epsilon}^*)|A_0|S_{B_c}(0)\rangle = \frac{-(E_{p_b} + m_b)(\hat{\epsilon}^* . \vec{k})}{\sqrt{(E_{p_b} + m_b)(E_{p_b+k} + m_{c/u})}}. \tag{A22}$$

Taking the above spin matrix elements, the expressions for hadronic amplitudes are compared and the model expressions for form factors, $V(q^2)$, $A_0(q^2)$, $A_+(q^2)$ and $A_-(q^2)$ are obtained as

$$V(q^2) = \frac{M + m}{2M}\sqrt{\frac{ME_k}{N_{B_c}(0)N_X(\vec{k})}} \int d\vec{p}_b \mathcal{G}_{B_c}(\vec{p}_b, -\vec{p}_b)\mathcal{G}_X(\vec{k} + \vec{p}_b, -\vec{p}_b),$$

$$\times \sqrt{\frac{(E_{p_b} + m_b)}{E_{p_b} E_{p_{c/u}}(E_{p_{c/u}} + m_{c/u})}}, \tag{A23}$$

$$A_0(q^2) = \frac{1}{(M - m)}\sqrt{\frac{Mm}{N_{B_c}(0)N_X(\vec{k})}} \int d\vec{p}_b \mathcal{G}_{B_c}(\vec{p}_b, -\vec{p}_b)\mathcal{G}_X(\vec{k} + \vec{p}_b, -\vec{p}_b),$$

$$\times \frac{(E_{p_b} + m_b)(E^0_{p_{c/u}} + m_{c/u}) - \frac{|\vec{p}_b|^2}{3}}{\sqrt{E_{p_b} E_{p_{c/u}}(E_{p_b} + m_b)(E_{p_{c/u}} + m_{c/u})}}, \tag{A24}$$

$$A_\pm(q^2) = \frac{-E_k(M + m)}{2M(M + 2E_k)}\left[T \mp \frac{3(M \mp E_k)}{(E_k^2 - m^2)}\{I - A_0(M - m)\}\right], \tag{A25}$$

with

$$E^0_{p_{c/u}} = \sqrt{|\vec{p}_{c/u}|^2 + m^2_{c/u}}, \quad T = J - (\frac{M - m}{E_k})A_0,$$

and

$$J = \sqrt{\frac{ME_k}{N_{B_c}(0)N_X(\vec{k})}} \int d\vec{p}_b \mathcal{G}_{B_c}(\vec{p}_b, -\vec{p}_b)\mathcal{G}_X(\vec{k}+\vec{p}_b, -\vec{p}_b),$$

$$\times \sqrt{\frac{(E_{p_b}+m_b)}{E_{p_b}E_{p_{c/u}}(E_{p_{c/u}}+m_{c/u})}}, \tag{A26}$$

$$I = \sqrt{\frac{ME_k}{N_{B_c}(0)N_X(\vec{k})}} \int d\vec{p}_b \mathcal{G}_{B_c}(\vec{p}_b, -\vec{p}_b)\mathcal{G}_X(\vec{k}+\vec{p}_b, -\vec{p}_b),$$

$$\times \left\{ \frac{(E_{p_b}+m_b)(E^0_{p_{c/u}}+m_{c/u}) - \frac{|\vec{p}_b|^2}{3}}{\sqrt{E_{p_b}E^0_{p_{c/u}}(E_{p_b}+m_b)(E^0_{p_{c/u}}+m_{c/u})}} \right\}. \tag{A27}$$

Therefore, the relevant form factors obtained in terms of model quantities, the helicity amplitudes, and the decay rates for $B_c \to \eta_c(J/\psi)l\bar{\nu}_l$ and $B_c \to D(D^*)l\bar{\nu}_l$ are evaluated, and our predictions of LFU ratios are listed in Tables 2 and 3.

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
