# Peer review of "A Light Shed on Lepton Flavor Universality in B Decays"

_universe, doi:10.3390/universe9030129_

Round 1

Reviewer 1 Report

Dear Authors,

I have read carefully your work and I have several comments most related on the structure of the draft. I also have few questions and suggestions, all summarised below.   - line 19: units —> blocks - in the introduction, fermions and quark are introduced.  I suggest to revisit the introduction adding also the quark flavors. While the leptons are described in details, nothing was introduced about quark flavours, which is also important for this work. - lines 21, 22: … generations having two members each, and the property which .. - refer to the PDG values for the electron, muon and tau masses -line 26: I don’t like the sentence “low mass” without a discussion on this aspect. I understand that this point is out of the scope of this review, but the upper limit on the neutrino masses based from experimental observation could be indicated (with the related references) - lines 32 and 33: maybe it is better to report the order of magnitude differences instead of the lifetimes - lines 33 and 34: it is true also for the quarks - line 38: add comma after “universal” - sentence at line 49: I don’t understand the link of this sentence with the rest of the text. Is the term “certainly” too much strong?  There are some theoretical models or references supporting this affirmation? - line 53: “ … boson W^\pm, while the …" - line 55: “ … weak boson (Z^0) at thee-level, or by virtual …" - line 58: “ … of the LFU has been ….” - Lines 58-74: I suggest to change this part by introducing first the oldest measurements, and concluding with the new one from LHCb - line 81: “accumulated” is not the right word. In these works a phenomenological analysis was done and loop effects inside the model are calculated and studied in order to explain the experimental observations. - Line 84; it is very important to define what R is. The ratio is never defined in the text - line 87: “Authors of Ref. [43]…" - caption of figure 1: LP —> LQ ? - line 93: remove “also” - lines 95: “In this context, we present our results … framework.” What do you mean at line 97? I suggest to remove the sentence at line 97 if it is related to the model description in section III. - line 114: remove “expectation” - lines 116, 120 : fig. —> Fig. - line 121: “…. FCC, i.e. ….” add comma - line 128 and 129: What do you mean here? Without any additional explanation and the introduction of Majorana fermions, this sentence could be misleading. I suggest to remove it.  - line 130: you should add the reference for the LHCb, BaBar and Belle analyses - lines 134 - 167: this part is very confused and must be re-written.  The first point is that LHCb is a pp experiment (7-8-13 TeV), while BaBar and Belle collected data from e+e- annihilation mainly at the Upsilon(4S) center of mass energy. This point should be highlighted.  As second point, I suggest to report the references of the LHCb, Belle and BaBar detectors instead of an incomplete description. - line 169: add comma after “quark” - line 170: “fb” not in italic - line 168: add here the reference of this LHCb analysis - line 182: remove “While" - line 183: add comma before “where"  - Table I: where come from the numbers reported in this table? The references are crucial here - figures: the style of the Feynman diagrams are different for all the figures, whereas it should be the same throughout the paper.  In figure 3 there also X1 and X2 which are not explained in the caption Caption figure 2(b): add B0 - line 196: BDbar all in math world  - line 211: up to now, QCD in never introduced. Maybe, it could be introduced in section I. - line 214: Numerous —> Several - line 220: add references for the mentioned models - line 231: “ where \gamma^0 is …., and V(r)=(ar^2_V_0). Here, r is the … antiquark, whereas a and V0 are …. using hadron spectroscopy [53]: … " - line 247: “The choice of the confining potential U(r) reported in Eq. (1) results simple and tractable when analysing various hadronic properties, and it provides  adequate tree-level ..." - line 253: are they momentum or four momentum? - line 276-277: “ … and spin, while \Lambda ….. “ - line 285: “… respectively, and m_l (M) is the mass of charged lepton (B_c meson).” - line 287: define what the observable R is. Note that the style should be consistent throughout the paper  - line 300: what is the clear experimental signature? - table II, caption: it is the ratio of BR - line 324: coefficientS ….. operatorS - Line 325: Authors of Ref. …. - line 326: add comma before “where” - line 333: parameter —> parameterS ? - line 337 and table IV: are “c” and “C” the same? If yes, use the same name throughout the paper - table IV: add also the references on the table. In the first column: Angular observables —> Observables Always in table IV, are the value of C and C’ exactly the same? - line 350: is there an error in the symbol? - lines 364-366: can you provide more details on the way the hadronic uncertainties were cancelled out? - line 383: …. authors of Ref. [92] have used charm penguins …. - line 385: “Additional details can be found in Ref. [93], in which the numerical comparison for the neutral mode …. was also done. “ - line 398: remove “the” - line 403: Authors of … - line 404: is it necessary to report the values of the Wilson coefficients C9 and C10? They can be removed - line 408-409: “Recently, the LBCb Collaboration ….” - line 415: what do you mean with the sentence “ which generated a good fit to the data” referred to ref. 99? - line 426: In this paper … - line 448: "... unprecedented precision “  Is there some studies on this point with the expected precision that can be reached after Run 3? 

Author Response

Q1 by the reviewer: "I have read carefully your work and I have several comments most related on the structure of the draft."

ReplyWe would like to thank the reviewer for the critical review and meticulous scrutiny of our manuscript, pointing out some shortcomings including obvious typos and grammatical errors, and giving useful suggestions for which the revised manuscript could be cast in the present form. We have gone through the article carefully, incorporated the referee's suggestions, and rectified them. With this, we hope our revised manuscript is ready for further processing.

Q2 by the reviewer: "Is there some studies on this point with the expected precision that can be reached after Run 3?"

ReplyYes, Run 3 data collections are expected to boom approximately three times larger in three years and will help in reducing the statistical and systematic uncertainties, and the predictions of world averages will be in agreement with SM predictions. Future hadron machines such as the FCC-hh: The Future Circular Collider (hadron-hadron) is presently under study which would extend the reach for direct observation of NP mediators into the multi-TeV range leading to unprecedented precision for flavor measurements. An indirect NP observation could also be possible at FCC-hh. A detailed study on upcoming Run 3 data analysis has been presented in Ref. Reviews of Modern Physics 94, no. 1 (2022): 015003.

Reviewer 2 Report

The authors calculate four LFUV observables in the relativistic independent quark model. I think their calculations are valuable additions to the field, hence this paper is worth publishing. The authors should take into account the following criticisms before I can recommend publication:

1. The paper currently reads like a review article that includes even a review of what fermions contribute to observables in the SM. A large portion of the paper is, therefore, narrative that obscures the actual novel results obtained by the authors. I suggested trimming such extraneous review to highlight novelty of the article itself.

2. Perhaps the authors are unaware -- the first analysis that pointed out that the ratio-type flavor anomalies (charged current and neutral current) can be unified under an effective field-theory framework was arXiv:1412.7164 [hep-ph]. Later on model analyses pinpointed the type of new interactions that could be responsible for unified frameworks for the flavor anomalies, for example in arXiv:1609.09078 [hep-ph] and arXiv:1806.07403 [hep-ph], among other articles. I suggest that the authors include these and similar citations in their article for the sake of completeness.

3. Recent LHCb measurement presented in arXiv:2212.09152 [hep-ex] supersedes previous RK, RK* measurements and the new data is now consistent with the Standard Model. This will clearly have an impact on the authors' narrative and I suggest that the authors reorient their narrative based on this very recent knowledge.

If the above comments are taken into account I will be willing to recommend this article for publication.

Author Response

Q1 by the referee: "The paper currently reads like a review article that includes even a review of what fermions contribute to observables in the SM. A large portion of the paper is, therefore, narrative that obscures the actual novel results obtained by the authors. I suggested trimming such extraneous review to highlight novelty of the article itself."

Reply: We thank the referee for taking the time in reviewing our manuscript and pointing out some shortcomings. We have gone through the remarks of the referee carefully and trimmed some extraneous sentences to highlight the novelty of the article.

Q2 by the referee: "Perhaps the authors are unaware -- the first analysis that pointed out that the ratio-type flavor anomalies (charged current and neutral current) can be unified under an effective field-theory framework was arXiv:1412.7164 [hep-ph]. Later on model analyses pinpointed the type of new interactions that could be responsible for unified frameworks for the flavor anomalies, for example in arXiv:1609.09078 [hep-ph] and arXiv:1806.07403 [hep-ph], among other articles. I suggest that the authors include these and similar citations in their article for the sake of completeness."

Reply: We thank the referee for directing us to the relevant references. We have included them in our revised manuscript to make our article more complete.

Q3 by the referee: "Recent LHCb measurement presented in arXiv:2212.09152 [hep-ex] supersedes previous RK, RK* measurements and the new data is now consistent with the Standard Model. This will clearly have an impact on the authors' narrative and I suggest that the authors reorient their narrative based on this very recent knowledge."

ReplyIn response to the above suggestions of the reviewer, we have reoriented the manuscript as per the recent measurements provided by LHCb and other collider experiments which are now consistent with the Standard Model predictions.

With this, we hope our revised article is worth publishing.

Reviewer 3 Report

The authors discuss the issue of Lepton Flavour Universality (LFU) in FCCCs and FCNCs in B decays, in the context of experimental measurements in the last decade.

Before collecting any specific comments to the manuscript, I think the authors should be more clear about the scope of this "review". In particular, the first thing to note is that the experimental status of LFU in b-->s transitions is completely outdated since the day after this paper was posted in the arxiv, and the authors did not have the sense to update it prior to sending it to the journal. The authors should address this problem first, then judge the scope of the paper, and then if the result is still meaningful, resubmit.

Author Response

Q1 by the reviewer: "The authors discuss the issue of Lepton Flavour Universality (LFU) in FCCCs and FCNCs in B decays, in the context of experimental measurements in the last decade.

Before collecting any specific comments to the manuscript, I think the authors should be more clear about the scope of this "review". In particular, the first thing to note is that the experimental status of LFU in b-->s transitions is completely outdated since the day after this paper was posted in the arxiv, and the authors did not have the sense to update it prior to sending it to the journal. The authors should address this problem first, then judge the scope of the paper, and then if the result is still meaningful, resubmit."

ReplyWe thank the referee for the comment. We have updated our manuscript as per the recent results by LHCb and several other collider experiments for b->s and b->c transitions and reoriented the manuscript as per the latest measurements which are now in precision with the Standard Model predictions. In the paper, we have calculated and reported our results of four LFUV observables in the relativistic independent quark model and we think our results are valuable additions to the field. In the absence of predicted data from established model approaches, in the literature, our predictions for LFUV observables for the charm and charmonium higher states can also be useful to identify the Bc channels in the upcoming Run 3 data at LHCb. Hence, we are certain that our revised manuscript is worth resubmitting.

Round 2

Reviewer 1 Report

Thanks a lot for this updated version. I found a lot of improvements. I still have additional minor comments summarised below:

table 1: Tau --> tau (all in not capital letters)

line 48: BaBaR --> BaBar

before eq. (1): "  .... BaBar Collaboration in 2012 [30]. In particular, the LFU ratio is defined as: ...., where l = e or mu, D^(*) refers to D (D^*) meson, and Br indicates the branching ratio. The results presented in Ref.[30] disagree with the SM at 3.4 sigma. These measurements ... "

line 65: .... thus imply LFUV. However ...

line 71: "fb" not in italic

line 75: ...(see Table 3). 

Equation 3: add comma at the end of equation

line 81: remove "where .... K*."

line 85: missing "-1" and "fb" not in italic

line 122: B-factories 

line 133: BR already defined as Br (line 56)

line 134: BR-->Br

line 138: do you mean "trillion of bbar pairs in 2020"?

line 141: the reference 47 maybe is wrong. I didn't fount in it the information reported in this sentence. In addition, you may leave a space between numbers and unit and use $\mu\rm{b}$ since "b" is not in italic.

lines 149-151: I still continue to don't understand this sentence. Didn't is more clear to simply something like "at LHCb all B-hadron species can be produced"? At the Upsilon(4S) c.m. energy (10.54 GeV), instead, only B0 or B+- meson pairs are allowed. Is it what do you want to highlight here

line 173: add dot at the end of the sentence

line 176: one order of 

line 186: ... Fig. 3. The study of ...

line 203: remove comma after "where"

after eq. 8: ... denote parent (Bc) and daughter (X) mesons, lepton and neutrino four-momentum, respectively.

line 215: functionS

after eq. 12: you can remove the \chi symbol since it is not present in any equation  

Table 6: last row, first column: tau --> tau^+ ? The table should be entered in the middle of the page

Before eq. 5: Recently, there has been ... showing deviation from SM predictions. Authors of Ref. [10] reported evidence of LFUV with a ....

Eq. 15: it is missed the statistical uncertainty. In addition, this result is from ref. 14 (2014), while references 10 and 52 are more recent (2019). Why not include the 2019 result?

line 305: it's --> it is

line 333: reported in Ref. [5]

line 334: .. on the Future Circular Hadron Collider FCC-hh at CERN ...

line 338: at Fermilab

line 340: collaboration --> Collaboration

line 343: In the LFUV ...

Author Response

We sincerely thank the referee for the comments. Please find below the mentioned changes which we have incorporated into the revised manuscript.

  1. We have incorporated all the grammatical comments in the revised manuscript raised by the referee.
  2. We have removed common knowledge of elementary particle physics to keep our manuscript more focused on our area of research and results, keeping in mind that the readers will have knowledge of elementary particle physics.
  3. Thank you for pointing out that Ref. 47 was wrong. We have corrected it with the proper reference.
  4. We have updated the results and equations. Also, added the appendices for more details about our framework for the sake of completeness.

Reviewer 2 Report

The authors have taken care of comments 2 and 3 from earlier. However, I had recommended that the authors trim the article to show its novelty. Instead, the authors added a lot of unnecessary information. I don't think it is necessary to quote lepton and quark masses. Furthermore, the authors can assume that their readers will likely know what lepton-flavor universality and its violation are -- no need to elaborate. 

I think the authors should carefully present only parts of the narrative absolutely necessary. After the authors have had that chance, I recommend this article for publication.

Author Response

We sincerely thank the referee for the comments. As per the suggestion, we have trimmed the article to show its novelty and removed the information on elementary particle physics.

We have kept only the parts required for the proper narrative of the article. We have also added appendices to present more details about our model for the sake of completeness.

We hope our revised manuscript is now suitable for publication.

Reviewer 3 Report

As a review, I find that this paper falls short in many respects. The first thing to note is the unbalance between the two types of observables considered: charged currents vs neutral currents. It would seem the authors lack the expertise to dive deeply into the FCNC case, while they feel more experienced in the CC case. They could review the CC case on its own, if they feel more comfortable. This unbalance leads to a considerable quality cut. But the review itself is deficient in many other aspects. One of the more important issues (especially in the charged current case) is the issue of form factors. The authors only skim through the subject, and they do so by extending on their RIQM model. However a good review should be thorough on this issue: HQET relations, light-cone sum rules, lattice calculations, momentum dependence and interpolation in different kinematic regions, etc. Tests of FNU in the charged sector are also deeply connected to FNU in the leptonic sector, where very precise tests have been performed (in particular with tau decays). There is also a large amount of related work on the angular distributions of semileptonic B-->D* decays, and other constraints from e.g. B_c decays. On the FCNC side, the R_K(*) anomalies were inextricably connected to the b-->smumu anomalies. Here there is also a lot to say about local and non-local form factors and hadronic uncertainties. The huge amount of literature on this issue is ignored in this review. While the R ratios are quite clean within the SM (where FNU is small), this is no longer the case in models with a large violation of LFU. The connection between R ratios in FCNCs and b-->smumu observables depends crucially on the quantity of LFU new physics. Nothing of this is discussed in any depth. This connection is even more motivated in combined explanations of b-->c and b-->s FNU, on which a HUGE literature exists, but completely ignored in this review. On top of this, there are many good reviews on these topics already written in the last few years, so the authors should motivate the need for this one. Taking into account its serious deficiencies, I do not see the motivation.

Thus my recommendation is: narrow down the scope of the review and be more thorough on the material covered. Unfortunately I think this is already a very major modification of the present review. 

Author Response

We sincerely thank the referee for the comments and understand the fact that our review does not cover broader areas, however, our goal is to present a review for a specific area where our model is applicable. Hence, we think that this article is worth publishing.

We request the referee to look at the attachment for our more detailed answer.
